# Study on the Spatiotemporal Evolution of the Ecological Landscape and Construction of an Ecological Network: A Case Study of Hebei Province

**Shijie Gu †, Shuhuan Li †, Shuai Wu, Bing Tian *, Yincui Hu, Minmin Cui and Mingze Sun**

Hebei Key Laboratory of Geological Resources and Environment Monitoring and Protection/Hebei Key Laboratory of Environmental Change and Ecological Construction/Hebei Technology Innovation Center for Remote Sensing Identification of Environmental Change, School of Geographical Sciences, Hebei Normal University, Shijiazhuang 050024, China; gushijie@stu.hebtu.edu.cn (S.G.); lishuhuan2020@163.com (S.L.); ws18732427638@163.com (S.W.); huyincui@hebtu.edu.cn (Y.H.); minm_cui@163.com (M.C.)

* Correspondence: tianbing@hebtu.edu.cn
† Contribute equally to the paper on behalf of the author.

**Abstract:** Research on the establishment of a regional ecological network can provide a scientific basis and valuable guidance for the protection of regional animals and plants, water conservation, sustainable resource utilization, and optimization of land use patterns. This study investigated the impacts of land use changes on the ecological security pattern using morphological spatial pattern analysis (MSPA), the minimum cumulative resistance (MCR) model, and the analysis of spatiotemporal changes and fragmentation of land use types. The results indicate that from 2000 to 2020, the dominant trend in land use types was the expansion of cultivated land, grass land, and wood land. Although the proportion of cultivated land was the largest and was concentrated in the southern part of Hebei Province, the total area of cultivated land showed a decline. Landscape index calculations revealed an increase in fragmentation of the overall landscape ecological patches, while the spatial and quantitative distribution of landscape types gradually became more uniform. Furthermore, 52 patches with the highest landscape index were identified as ecological sources, mainly located in northern Hebei Province in 2020. MSPA calculations showed that elevation, slope, and land use type contributed significantly to the comprehensive resistance surface. Using the MCR model, an ecological network for Hebei Province was constructed, consisting of 114 ecological corridors and 28 ecological nodes. The ecological corridors exhibited a distribution pattern of high density in the north and low density in the south, while the ecological nodes enhanced overall ecological connectivity in the region. Based on the current ecological environment, it is recommended to increase the number of ecological corridors and ecological nodes to enhance ecosystem stability.

**Keywords:** morphological spatial pattern analysis (MSPA); minimum cumulative resistance (MCR); land use type; ecological corridor; ecological node

## 1. Introduction

The ecological environment is the basic condition for human survival, production, and life. However, due to the needs of China's early economic development, urban construction mainly focuses on the development of industrial economy, the landscape ecological pattern planning is less, and the living environment continues to fragment [1,2]. The increase in industrial development and urban infrastructure construction occupies a large amount of ecological space. And then, the development of cities and towns leads to great changes in regional land use types, and change of local landscape pattern destroys the ecological environment of the original biological species, gradually separating ecological corridors from ecological patches, causing serious fragmentation of the ecological patches

and threatening biodiversity [3–5]. However, with the continuous advancement of ecological civilization construction and environmental protection, ecological security issues have gradually improved in recent years. Facing the current ecological environment, it is significant to understand the actual status and dynamic change characteristics of regional natural resources for the management of resources and sustainable development of the ecological economy [6–8].

Landscape pattern is the arrangement and combination of ecological elements of different shapes and sizes at spatial scales. It is easy to be affected by natural and social factors, among which land use/cover change is one of the most important factors affecting its structure, function, and dynamics [9,10]. At present, most research is aimed at ecological security. Yang et al. [11] used dynamic attitude and transfer matrix of land use/cover, as well as landscape pattern indices, to influence the landscape pattern of the Yangtze River Basin. The changes caused by human activities were the main driving factors of land use/cover. Wang et al. [12] used the landscape index to assess the impact of land cover after the earthquake in Sichuan Province on a micro scale, and the results showed that ecological recovery and subsequent development of the study area after the earthquake were good. Echeverría et al. [13] used an integrative analysis to investigate the impacts of landscape change on local land cover and the resulting distinctive landscape patterns in the southern region of Chile, showing that changes in the local landscape pattern created three phases, and land cover changed in different spatial patterns according to the landscape phases.

The ecological network is composed of the ecological source area and ecological corridor, where the ecological source area is the source point of biological species and energy diffusion, and the ecological corridor provides the channel of biological and energy flow. The ecological corridor originated from the concept of "Greenway" put forward by Little [14] in 1990. Fabos [15] summarized the functions of greenway into three aspects, namely, the ecological corridor, leisure corridor, and historical or natural protection corridor. At present, the model of "ecological sources–ecological corridors–ecological nodes" is adopted to construct the ecological network [16]. There are different opinions on the ways to identify ecological source areas and ecological corridors [17]. The methods of selecting ecological source areas include identifying the land use types of key areas [18–20] and selecting source areas by using a landscape pattern index [21,22]. Morphological spatial pattern analysis (MSPA) is an image processing method that uses mathematical morphological operation theories such as corrosion, expansion, open operation, and close operation to segment, identify, and classify raster images, etc. Later, it was widely used in landscape ecology research [23–25]. The minimum cumulative resistance (MCR) model is used to calculate the ecological species from source to the destination cost price model [26,27]. In general, the method of using MSPA to measure landscape structure and MCR to extract the ecological corridor is more widely used [17]. Hu et al. [28] comprehensively evaluated the ecological network space of Wuhan City with an MSPA-MCR model and quantitatively analyzed the importance of ecological corridor protection by combining the gravity model. Wei et al. [29] constructed an ecological security pattern of the Ebinur Lake Basin. Based on the MSPA-MCR model and combining with the large landscape connectivity index and the probability of connectivity values, Dai et al. [30] established a comprehensive ecological security network evaluation method for the urban agglomeration around Poyang Lake based on the MCR model and the Duranton and Overman Index and supplemented the industrial agglomeration theory to form a dual evaluation system of economics and landscape ecology. Kang et al. [31] used MSPA and the network analysis model to analyze the forest network structure in North Korea by classifying the forest cover into different types, identifying the key core and bridging areas of the forest region and providing scientific guidance and management strategies for local forest management and protection. In general, few relevant studies have combined the evolution of landscape pattern derived from the land use type transformation matrix with the current MSPA-MCR model to construct an ecological security pattern and assisted ecological network construction through the landscape pattern index.

In this study, based on the data of land use types in Hebei Province, the transition matrix and seven landscape indices were used to explore the spatiotemporal evolution and distribution characteristics of landscape patterns. And then, MSPA was used to identify and classify the distribution characteristics of land use types and related ecological patches. Ecological patches are combined and update with the data of ecological protected areas. Delta values for the PC (dPC) index were selected to analyze the connectivity of landscape ecological patches, and the patches with good connectivity were used as an ecological source area for constructing ecological networks. Finally, the MCR model was used to construct the ecological network of the study area, seven types of main resistance factors were selected to determine the factor weights by the analytic hierarchy process (AHP) method, and the resistance evaluation index system was constructed. The comprehensive resistance surface of the study area was formed by weighted superposition. The ecological corridor and ecological node were identified by the potential least-cost paths (LCPs) method, and the landscape ecological network of Hebei Province was constructed. In short, changes in land use types can directly impact the ecological security pattern, while the construction of ecological security patterns can guide the rational planning and management of land use to some extent. Their coordination and rational planning can contribute to maintaining ecosystem stability and biodiversity, as well as enhancing societal ecological security. Therefore, when planning land use and designing ecological security patterns, it is necessary to consider their relationship comprehensively in order to achieve sustainable land use and ecosystem management. The research results show the theoretical and technical basis for ecosystem protection and the rational development and utilization of resources in Hebei Province and other provinces, as well as provide a reference for the evolution of land use types and the construction of the ecological security pattern.

## 2. Study Area and Analysis Method

### 2.1. Study Area

Hebei Province (36–43° N, 113–120° E) is located in the northern part of the North China Plain, which is surrounding Beijing City and Tianjin City, and it consists of 11 prefecture-level cities with a total area of $1.888 \times 10^5$ km$^2$. The terrain is inclined from northwest to southeast; the northwest area is mainly mountainous, and the southeast area is mainly plains. It is the only province in China with plateaus, plains, mountains, hills, lakes, and beaches. It has a typical temperate semi-humid and semi-arid continental monsoon climate, with annual precipitation of 200–700 mm and annual average temperature of −2~15 °C [32,33]. In recent years, due to the overdevelopment and irrational use of land, environmental problems have become severe. Figure 1 reports an overview of the study area.

### 2.2. Data

The land cover type data v2000, v2010, and v2020 used in this study come from the GlobeLand30 dataset (http://www.globallandcover.com/ accessed on 27 December 2022) provided by the National Geographic Center of China, with a spatial resolution of 30 m [34]. According to the GlobeLand30 data classification system and research needs, landscape types were divided into 7 categories, namely, cultivated land, wood land, grass land, shrub land, wet land, water, building land, and unused land.

The Normalized Difference Vegetation Index (NDVI) dataset collected in this study comes from the National Aeronautics and Space Administration (NASA, https://ladsweb.modaps.eosdis.nasa.gov/ accessed on 7 March 2023) [35]. The MOD13 Q1 data product was used, with spatial and temporal resolutions of 250 m and 16d, respectively. The DEM data used were derived from SRTMDEM data with 90 m resolution of Geospatial Data Cloud (https://www.gscloud.cn/ accessed on 13 January 2023). By preprocessing the original DEM data, such as Mosaic, clipping, projection transformation and unified coordinate system, elevation, and slope information were extracted. In addition, the

road and river data in Hebei Province came from the OpenStreetMap website (https://www.openstreetmap.org/ accessed on 25 March 2023) [36].

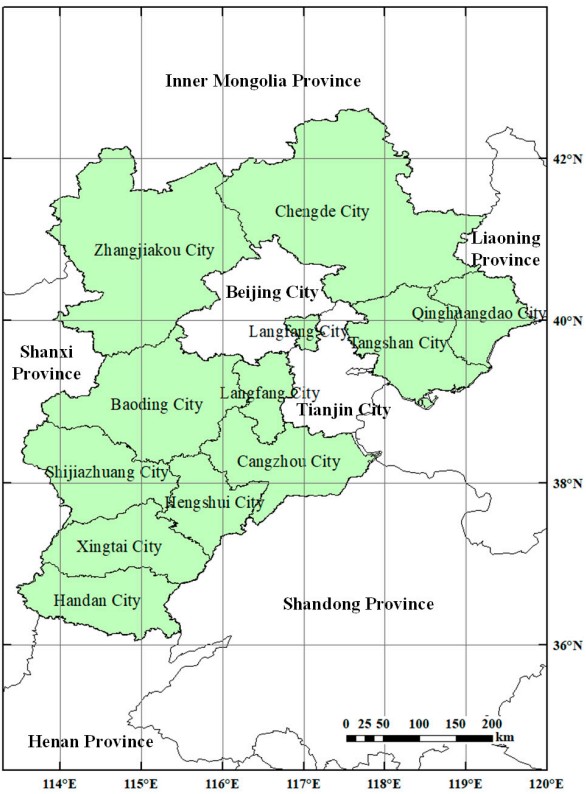

**Figure 1.** Study area profile.

### 2.3. Analysis Method

2.3.1. Landscape Type Transfer Matrix

The landscape type transfer matrix can intuitively reflect the transformation of various landscape types (conversion area of each landscape type) during the study period. Based on land use data, the statistical and spatial changes of landscape types in the study area can be analyzed [26,37]. The formula of the transfer matrix is as follows:

$$U_{i,j} = \begin{bmatrix} U_{11} & \cdots & U_{1m} \\ \vdots & \ddots & \vdots \\ U_{m1} & \cdots & U_{mm} \end{bmatrix} \tag{1}$$

where U is the area of a certain land use type; i and j represent the beginning and ending land use types during the study period, respectively; and m denotes the number of land use patches.

2.3.2. Landscape Index Analysis

Landscape index reflects the composition, structure, and spatial distribution characteristics of the landscape pattern in digital form. The landscape pattern reflects the actual spatial distribution of each landscape type and determines the spatial heterogeneity of the landscape, affecting the ecological process of the region. Based on the study on the spatiotemporal evolution of the landscape pattern and the subsequent construction of the ecological network, we selected 7 landscape indexes from the scales of patch type and landscape type to describe the dynamic changes of the landscape pattern in the study area from the aspects of landscape quantity, structure, and form [38,39]. Fragstats4.2 software was used to calculate the landscape index at the patch level. The 7 landscape indexes were Patch Density (PD), Largest Patch Index (LPI), Landscape Shape Index (LSI), Aggregation

Index (AI), Splitting Index (SPLIT), Shannon's Diversity Index (SHDI), and Shannon's Evenness Index (SHEI). Table 1 and Formulas (2)–(8) show the definition and calculation method of the landscape index.

$$PD = n_i / A \tag{2}$$

$$LPI = \frac{\max\limits_{j=1}^{n} a_{i,j}}{A} \times 100\% \tag{3}$$

$$LSI = 0.25 \sum_{k=1}^{m} e_{ik}' / \sqrt{A} \tag{4}$$

$$AI = \left[ \frac{g_{ii}}{max \rightarrow g_{ii}} \right] \times 100 \tag{5}$$

$$SPLIT = A^2 / \sum_{j=1}^{n} a_{i,j}^2 \tag{6}$$

$$SHDI = -\sum_{i=1}^{n} P_i In P_i \tag{7}$$

$$SHEI = \frac{-\sum_{i=1}^{n} (P_i In P_i)}{In m} \tag{8}$$

where $n$ is the total number of patches of a certain type; $a_{i,j}$ reports the area of each patch; $A$ is landscape area; $m$ stands for number of patches; $e_{ik}'$ denotes the landscape length of a certain type of patch, including the entire landscape boundary and all background edges; $g_{ii}$ represents the number of similar connections between patch type pixels based on a single counting method; and $P_i$ represents the area ratio of color patch area i to the entire landscape.

**Table 1.** Descriptive statistical analysis of 7 landscape indexes.

| | Landscape Index | Significance | Unit |
|---|---|---|---|
| Patch metrics | Patch Density (PD) | Representing the degree of differentiation of a landscape type (degree of landscape fragmentation). | ind·m$^{-2}$ |
| | Largest Patch Index (LPI) | Distinguishing the dominant type of a landscape. | % |
| | Landscape Shape Index (LSI) | Representing the complexity of the landscape shape. | - |
| | Aggregation Index (AI) | Describing the degree of aggregation of the patches. | % |
| Landscape metrics | Splitting Index (SPLIT) | Representing the degree of fragmentation of the landscape. A higher value means more fragmentation of a landscape. | - |
| | Shannon's Diversity Index (SHDI) | Reflecting landscape heterogeneity. Representing the complexity of landscape pattern composition. | - |
| | Shannon's Evenness Index (SHEI) | Representing the degree to which different landscape types are uniform in their number or area. | [0, 1] |

### 2.3.3. Morphological Spatial Pattern Analysis

MSPA is based on the principles of computer imaging and uses mathematical methods for recognition and segment grids at the pixel level, completing landscape patch calculations [17]. In this study, GuidosToolbox3.0 software was used to conduct MSPA on the binarization data of land use types in Hebei Province in 2020. Ecological land was taken as the foreground and non-ecological land as the background. It was divided into 7 landscape types: core, islet, perforation, edge, loop, bridge, and branch area. Table 2 provides the significance and area proportion of each type in the MSPA. The ratio is the proportion of the ecological land area.

**Table 2.** Descriptive statistical analysis of 7 landscape types with MSPA.

| Landscape Type | Significance | Area (km$^2$) | Ratio (%) |
|---|---|---|---|
| Core | Large ecological patches in landscape types, such as forest parks, nature reserves, etc. It is generally a biological habitat, playing an important role in maintaining habitat stability and biodiversity, and it can be used as an ecological source. | 51,690 | 27.50 |
| Islet | A small, isolated, fragmented patch. Generally, it refers to small urban park green space. | 1333 | 0.71 |
| Perforation | A gap formed when the core area is disturbed and destroyed, resulting in vegetation degradation. | 2695 | 1.00 |
| Edge | The transition area between the core area and the non-core area, which protects the core area from external interference and has an edge effect. It is generally a protected forest belt in a nature reserve. | 10,297 | 5.48 |
| Loop | Connecting corridors inside the same core area; when the patch area is large and the inner edge distance is too long, the loop area can ensure the energy exchange and material flow of the patch. It is usually a green belt or nature reserve. | 2406 | 1.28 |
| Bridge | A channel connecting two or more core areas has a ribbon structure. Generally, it is an ecological corridor that promotes the protection of ecology and the flow of matter and energy. | 2360 | 1.26 |
| Branch | Only one end is connected to the bridge area, loop area, edge area, or perforation area, and the other end is connected to the background area. | 3356 | 1.79 |

2.3.4. Ecological Corridor Construction Based on the MCR Model

The MCR model was used to build the ecological network of the study area, set the resistance value, and construct the buffer zone from 7 resistance factors including natural and social factors, and then the AHP was used to determine the factor weights and construct the resistance evaluation index system [40]. A comprehensive resistance surface of the study area was obtained by weighted superposition. Using LCPs, ecological corridors and nodes in the study area were identified according to ecological source points and comprehensive resistance surface, and a landscape ecological network composed of source points, corridors, and nodes in Hebei Province was constructed. The formula is as follows:

$$MCR = f_{min} \sum_{i=n}^{i=m} D_{i,j} \times R_i \tag{9}$$

where *MCR* represents the minimum cumulative resistance value; $D_{i,j}$ reports the spatial distance from ecological source point I to j; $R_i$ is the resistance coefficient of source point expansion; and $f_{min}$ provides the positive correlation function of the relationship between reactions *MCR*, $D_{i,j}$, and $R_i$.

**3. Results**

*3.1. Landscape Type Change Analysis Results*

Under the driving of natural and human factors, the area, structure, and mode of land use changed. Landscape pattern refers to the type, number, spatial distribution, and configuration of landscape component units, and it is the spatial structure characteristic of landscape pattern. Land use type is the most intuitive manifestation of landscape pattern change, and land use change reflects landscape change in spatial distribution. Therefore, understanding land use type change is helpful to the development of the regional landscape pattern [41,42].

Figure 2 provides the spatiotemporal distribution characteristics and area changes of land use types obtained using land use data from 2000, 2010, and 2020. Compared with 2000, the cultivated land area decreased by 9278.41 km$^2$ (4.94%) in 2020, but it was still the main land use type in the study area. The area of water and wet land also decreased, but the changes were relatively small. Under the influence of urban development, the area of building land continued to increase by 8828.79 km$^2$, followed by grass land area, which

increased by 673.54 km$^2$, and then the area of wood land increased by 230.68 km$^2$, with the area of shrub land and unused land increasing but the change being relatively small.

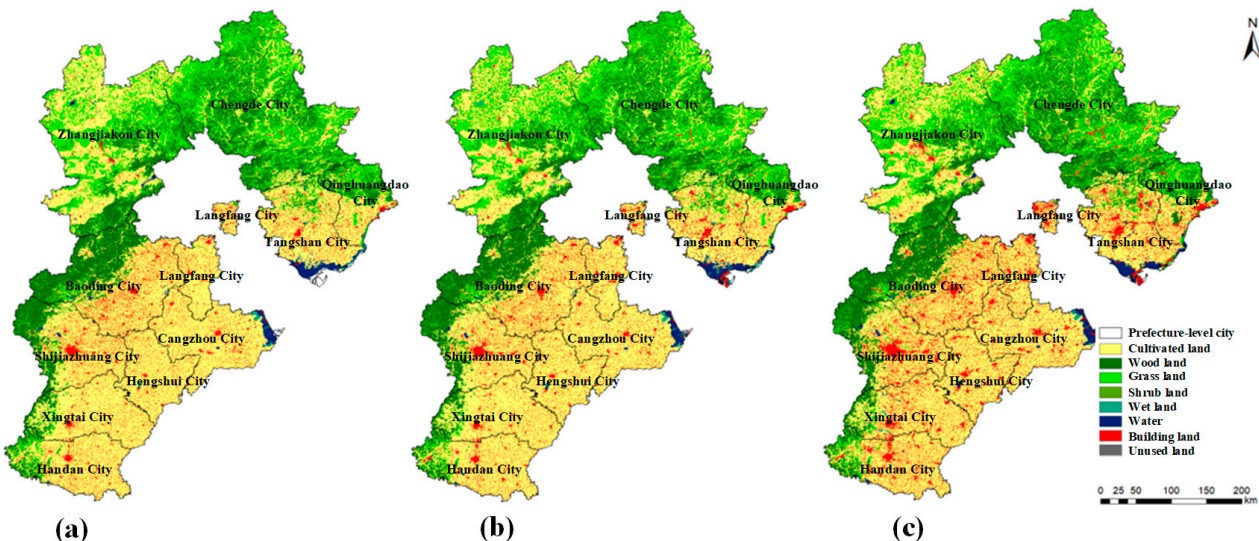

**Figure 2.** Spatiotemporal distribution of land use types in the study area (**a**–**c**) representing 2000, 2010, and 2020, respectively.

The landscape type transfer matrix can reflect the current situation and change the rule of objective things. Table 3 shows the changes in land use types from 2000 to 2020, with rows and columns representing the area of types in the starting and ending years, respectively. The result reports that the total amount of grass land and wood land did not fluctuate greatly due to the mutual transformation between grass land and wood land, but the structural layout of land use type changed. Building land was the largest increase in all land types, in which the conversion of cultivated land was the main one, and the rapid growth of population size and the improvement of urbanization level led to the corresponding expansion of urban land area. And then, the wet land area decreased by a total of 22.85 km$^2$, with conversion to water becoming the main factor of change. Shrub land area increased, and conversion to grass land was dominant. The area of water body was reduced by 27.72 km$^2$, with a large transformation to cultivated land, building land, and grass land. Finally, the main source of the growth of the unused land area was grass land, which was mainly due to the relationship between natural and social factors, leading to soil salinization and sandiness, increasing the area of unused land, while the conversion of unused land to other land types was relatively small.

**Table 3.** Land use type change based on transfer matrix.

| | | Year: 2000 | | | | | | | | |
|---|---|---|---|---|---|---|---|---|---|---|
| Unit: (km$^2$) | | Cultivated Land | Wood Land | Grass Land | Shrub Land | Wet Land | Water | Building Land | Unused Land | Total |
| | Cultivated land | 88,039 | 1106 | 2819 | 17 | 34 | 349 | 2097 | 7 | 94,469 |
| | Wood land | 1360 | 27,458 | 5689 | 50 | 3 | 16 | 30 | 2 | 34,607 |
| | Grass land | 4249 | 5556 | 25,961 | 113 | 37 | 105 | 82 | 28 | 36,130 |
| | Shrub land | 30 | 79 | 111 | 17 | 0 | 1 | 1 | 0 | 238 |
| Year: 2020 | Wet land | 111 | 5 | 51 | 0 | 92 | 38 | 2 | 0 | 299 |
| | Water | 507 | 33 | 99 | 1 | 155 | 1581 | 29 | 1 | 2405 |
| | Building land | 9412 | 112 | 609 | 2 | 25 | 143 | 8762 | 4 | 19,070 |
| | Unused land | 58 | 24 | 111 | 1 | 1 | 2 | 4 | 20 | 221 |
| | Total | 103,764 | 34,371 | 35,451 | 202 | 347 | 2234 | 11,007 | 62 | 187,439 |

### 3.2. Landscape Index Analysis Results

The landscape index has special significance in landscape ecology, and its size and change reflect the spatial changes and distribution characteristics of various landscape

types in a region [43]. Using Fragstats 4.2 software, the landscape index of the land use grid data in 2000, 2010, and 2020 were compared and analyzed in the study area. Table 4 shows the changes of landscape index at patch metrics.

**Table 4.** Landscape pattern index of Hebei Province in 2000, 2010, and 2020 (Patch Metrics).

| Land Use Type | Year | PD | LPI | LSI | AI |
|---|---|---|---|---|---|
| Cultivated land | 2000 | 0.0585 | 32.9833 | 165.7424 | 89.7014 |
| | 2010 | 0.0662 | 32.6826 | 174.9275 | 88.9790 |
| | 2020 | 0.0735 | 29.9267 | 198.6736 | 87.0469 |
| Wood land | 2000 | 0.1787 | 6.0018 | 262.5606 | 71.7549 |
| | 2010 | 0.1844 | 5.9982 | 261.0839 | 71.9633 |
| | 2020 | 0.1806 | 5.8606 | 258.9038 | 72.2242 |
| Grass land | 2000 | 0.2166 | 3.1258 | 326.2443 | 65.5424 |
| | 2010 | 0.2104 | 3.3895 | 322.9125 | 66.4049 |
| | 2020 | 0.2083 | 2.7697 | 321.0682 | 66.3744 |
| Shrub land | 2000 | 0.0264 | 0.0002 | 72.5294 | 4.3942 |
| | 2010 | 0.0286 | 0.0141 | 72.9763 | 13.7488 |
| | 2020 | 0.0273 | 0.0113 | 70.8485 | 13.4175 |
| Wet land | 2000 | 0.0024 | 0.0255 | 23.6243 | 75.7939 |
| | 2010 | 0.0025 | 0.0313 | 26.6778 | 78.1889 |
| | 2020 | 0.0044 | 0.0103 | 31.9007 | 58.5060 |
| Water | 2000 | 0.0259 | 0.4257 | 47.1375 | 80.6605 |
| | 2010 | 0.0256 | 0.4483 | 49.6944 | 79.5992 |
| | 2020 | 0.0020 | 0.0248 | 24.8689 | 73.4064 |
| Building land | 2000 | 0.1695 | 0.1318 | 203.4347 | 62.9309 |
| | 2010 | 0.1882 | 0.1927 | 207.7595 | 64.3880 |
| | 2020 | 0.2060 | 0.2918 | 229.4793 | 67.8470 |
| Unused land | 2000 | 0.0043 | 0.0022 | 30.7143 | 27.0175 |
| | 2010 | 0.0044 | 0.002 | 30.8353 | 27.7699 |
| | 2020 | 0.0257 | 0.2857 | 54.3333 | 78.9360 |

Firstly, the PD value reflects the degree of fragmentation. Among all landscape types, the value of grass land was the largest, indicating that which had the highest degree of fragmentation, followed by wood land and building land. Moreover, patch density gradually increased, indicating that part of the wood land and grass land were gradually degraded, and the degree of land fragmentation increased. However, the PD value of shrub land, cultivated land, unused land, water, and wet land were less than 0.1, and fragmentation degree was relatively small, indicating that these landscape types had small changes and were more concentrated.

Secondly, the LPI value reflects the dominant landscape types in study area. The LPI value of cultivated land was the largest, providing characteristics of dominant landscape type and showing a small decline trend with grass land and wood land. Water, building land, wetland, shrub land, and unused land values were both less than 1%. It can be seen that cultivated land, wood land, and grass land were the main landscape types in the study area. From the view of ecological protection, water, wet land, and shrub land areas should be expanded.

Thirdly, LSI value indicates the complexity of the landscape shape. Grass land and wood land value were relatively high and showed a downward trend, reflecting the irregularity and tendency towards regularity of these two landscape types. The LSI value of building land and cultivated land increased. Shrub land, unused land, water, and wet land value were relatively low, and the amplitude of change was relatively small.

Fourthly, AI value reflects the spatial distribution of plaques, and a higher value indicates a greater degree of plaque aggregation. AI values of landscape types in the study area from large to small were cultivated land, water, wet land, wood land, grass land, building land, unused land, and shrub land. The area change of landscape types will also affect the AI value of landscape types to a certain extent. For example, unused land and shrub land areas were the smallest in all landscape types, and the degrees of aggregation were also the smallest in all landscape types.

Table 5 shows the changes of the landscape index at landscape metrics. First of all, from 2000 to 2020, the SPLIT value increased from 7.9729 to 9.9171, and the degree of landscape separation was proportional to landscape fragmentation degree, indicating that the landscape fragmentation degree in study area gradually increased. Secondly, the SHDI value, as an indicator of ecological diversity, increased by 0.0952, reporting that landscape types in study area were becoming increasingly diverse. Finally, the SHEI value increased from 0.5818 to 0.6276, indicating that landscape dominance was low and all landscape types were evenly distributed.

**Table 5.** Landscape pattern index of Hebei Province in 2000, 2010, and 2020 (landscape metrics).

| Landscape Type Index | 2000 | 2010 | 2020 |
|---|---|---|---|
| SPLIT | 7.9729 | 8.3892 | 9.9171 |
| SHDI | 1.2098 | 1.2407 | 1.305 |
| SHEI | 0.5818 | 0.5967 | 0.6276 |

In summary, after comparing and analyzing the land use data in the study area for the years 2000, 2010, and 2020, we observed that the grass land, wood land, and building land had relatively high PD values, indicating high landscape fragmentation, low aggregation, and weak connectivity. The cultivated land, on the other hand, exhibited the highest LPI value, suggesting that it is the dominant landscape type in study area. Among all landscapes, grass land, wood land, building land, and cultivated land exhibited relatively high LSI values, indicating a high level of complexity in their shape. Additionally, there was an increase in AI values for wood land, unused land, and water, indicating a shift from scattered distribution to aggregated distribution. The increase in SPLIT value indicated growing fragmentation. Both SHDI and SHEI values also increased, indicating a shift towards a more uniform quantity and spatial distribution of landscape types in the study area. This change signifies a decrease in dominance by one or a few landscape types, resulting in increased stability of the landscape ecosystem.

*3.3. Ecological Source Area Study*

In order to further understand the impact of land use change and landscape pattern change on the current ecological security pattern, Figure 3 reports the Hebei Province landscape type distribution in the MSPA in 2020. The result shows that the core area was the largest among the MSPA landscape types, mainly concentrated in the northern, central, and southern parts of the study area, but the landscape patches were scattered and distributed in a band along the border of Hebei Province. The core areas of the southwest an in Langfang and Tangshan City had a few patches, and it was difficult to maintain ecosystem connectivity. The second largest area was the edge perforation area, accounting for 6.5% of all landscape types, both of which were the transition zone between the core and non-core area. And then, the bridge and loop area were the corridors connecting the core area, with an area of 4766 km$^2$ (2.54%), playing a positive role in maintaining the ecosystem. However, the branch area was referred to as the interruption of the ecological corridors. The connectivity of the islet areas was relatively poor. The two account for a smaller proportion. In summary, according to the landscape classification results, the core area had the largest area and the highest aggregation degree among all landscape types, and its connectivity was relatively good. Therefore, this study extracted the core area as a potential ecological source.

Based on the calculation results of the core area extraction by MSPA, there were 6414 patches of different sizes in the study area. The area of the patch for the surrounding ecological area impacted the range greater, and the connectivity was relatively high. In the process of selecting ecological sources, patches that were far away from the overall patch, scattered and small in area, were removed, and some small patches were merged with large patches. A total of 64 patches with large area, wide radiation range, and good connectivity were selected. And then, local ecological reserves had a high ecosystem service function, so

data of 17 nature reserves with vector range were selected as the source area. The data of 64 ecological patches were updated and combined, and finally, 62 ecological patches were generated. In the end, the importance of the core ecological patches identified by the MSPA method were unable to be distinguished. Through landscape connectivity evaluation, the connectivity between landscape patches can be judged. Therefore, this study used Conefor sensinode 2.6 software and dPC to calculate and evaluate patch importance [44]. Combined with relevant research [45,46] results and the current situation of the study area, the thresholds were selected as 100 m, 500 m, 1000 m, 2000 m, 3000 m, and 5000 m. The number and area of patch connections under different thresholds were compared. It was found that the importance of patches can be better reflected when the distance threshold was 2000 m. Therefore, 2000 m was selected as the distance threshold with a probability of 0.5, and the importance degree of ecological patches was evaluated based on dPC. In conclusion, based on the above analysis results, 52 representative ecological patches were selected as ecological sources for constructing the ecological network. Figure 4 provides the spatial distribution of the ecological source area in Hebei Province.

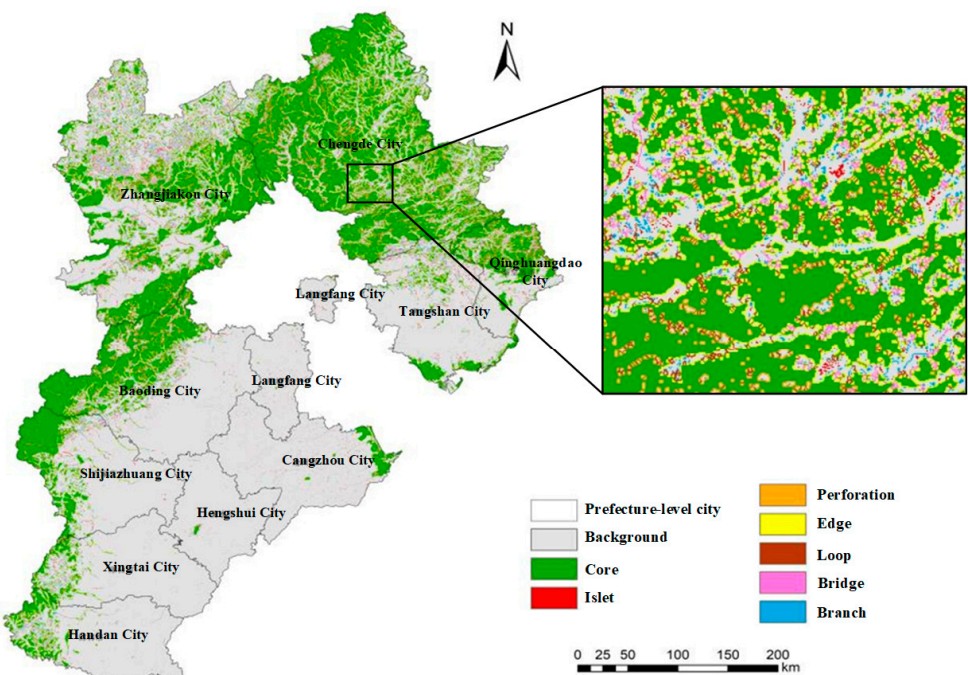

**Figure 3.** Classification of landscape types in Hebei Province based on MPSA.

### 3.4. Ecological Corridor Construction

The migration of biological species between sources is affected by landscape resistance. Different landscape types and internal structures of landscape types will lead to differences in the resistance that biological species need to overcome when moving from the origin source to the target source. The size of the resistance value depends on the degree of interference of the natural environment and social factors. In this study, seven evaluation factors were selected as resistance factors [47,48]. Natural factors included elevation, slope, land use type, and fractional vegetation cover (FVC). And social factors, river distance, rail/highway buffer zone, and Grade I road buffer zone were selected. The resistance value was uniformly set between 1 and 5, where 1 is the minimum resistance value, indicating that the biological species can reach the target source smoothly, and 5 is the maximum resistance value, indicating that biological species need to overcome the maximum resistance.

According to nature factor, first of all, Figure 5a,b shows the spatial distribution of elevation and slope resistance values. The change of elevation and slope had a certain degree of influence on vegetation growth, vegetation type, species migration, and ecological source area extension. Elevation and slope value are proportional to resistance. And then,

Figure 5c provides the spatial distribution of the land use type resistance value. The ecological source area was the area with higher ecological quality in the study area. The closer the land use type and ecological source, the less resistance. In the end, Figure 5d reports the spatial distribution of the FVC resistance value. FVC reflects the growth state and spatial distribution density of surface vegetation, and it is closely related to land use type and regional ecological protection, so it can be used as an important index to evaluate landscape ecology.

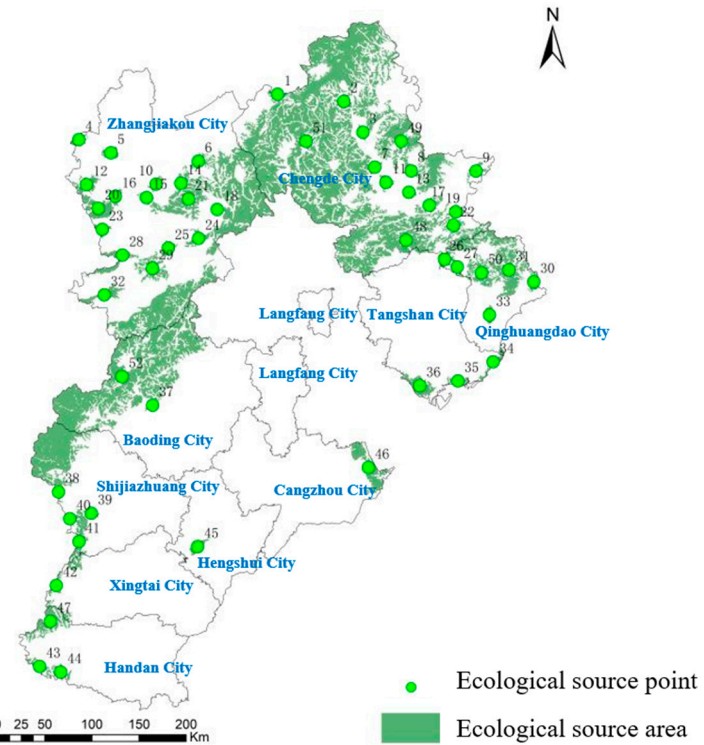

**Figure 4.** Spatial distribution of the ecological source area in Hebei Province.

According to the social factor, Figure 5e shows the river distance buffer zone. Rivers have a high ecological service function, providing convenient conditions for the migration of biological species, the maintenance of biodiversity, and the diffusion of ecological sources. Therefore, the distance of a river is inversely proportional to the size of resistance. And then, Figure 5f,g reports the rail/highway buffer zone and Grade I road buffer zone. Social development largely depends on traffic conditions. The improvement of a road network is conducive to regional development and construction, and this will also promote the development and utilization of the surrounding land, affecting the change of the regional landscape pattern. Therefore, it has a certain hindrance to the migration of biological species and the expansion of ecological sources.

After determining resistance classification and the resistance value, it is necessary to calculate the relative influence degree of resistance of each influence factor during the expansion of the ecological source and the migration of species, that is, the resistance factor weight. Table 6 shows the resistance factor evaluation information. Figure 5h reports the comprehensive resistance surface based on the calculated weight.

Using ecological source points and the MCR model to construct the ecological network can better show the cost of source points under the influence of various resistance factors. Using the ArcGIS 10.8 software distance analysis tool, a source point was taken as the ecological source, with other source points as the target source. The minimum cost path from each source point to other i-1 target sources was obtained. Through calculation, 531 ecological corridors were obtained from 52 source points. Figure 6a provides the ecological source point and ecological corridor spatial distribution characteristics. In terms

of spatial distribution, it can be seen that the corridors were mostly distributed in Chengde City, Zhangjiakou City, and Qinhuangdao City, which were dominated by mountainous and hilly landforms. When the ecological corridors were distributed in low mountains and hilly areas, the migration and diffusion of biological species were easier. When distributed in the high mountain area, the migration and diffusion were more difficult. The southern part of Hebei Province was mostly plains, mainly cultivated land, which was greatly disturbed by social factors, and the migration and diffusion of biological species will also be hindered.

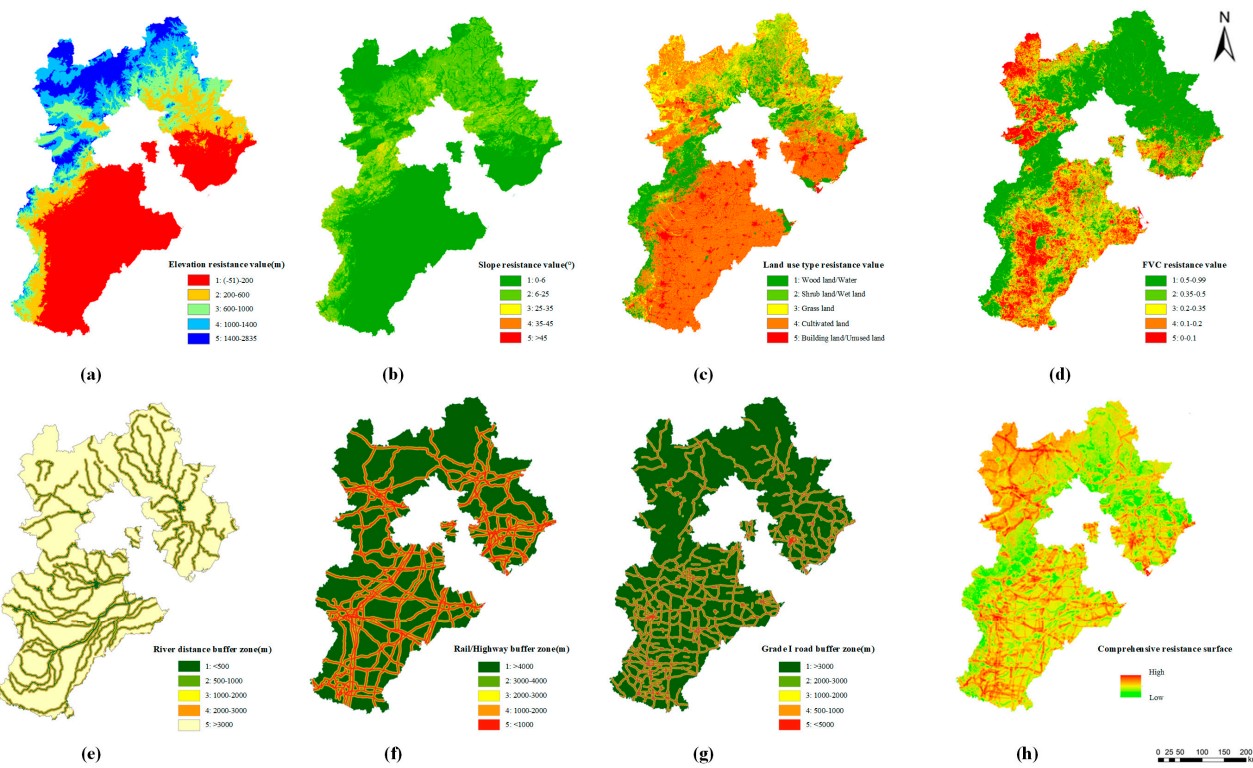

**Figure 5.** Spatial distribution of resistance classification (**a**–**h**) representing elevation resistance, slope resistance, land use type resistance, FVC resistance, river distance buffer zone, rail/highway buffer zone, Grade I road buffer zone, and comprehensive resistance surface, respectively.

**Table 6.** Weight evaluation of 7 resistance factors.

| | Resistance Classification | Resistance Value | Resistance Classification | Resistance Value | Resistance Classification | Resistance Value | Resistance Classification | Resistance Value |
|---|---|---|---|---|---|---|---|---|
| | Elevation (weight:0.1851) | | Slope (weight: 0.1784) | | Land use type (weight: 0.1594) | | FVC (weight: 0.1495) | |
| | −51–200 m | 1 | 0–6° | 1 | Wood land and water | 1 | >0.75 | 1 |
| | 200–600 m | 2 | 6–25° | 2 | Wet land and shrub land | 2 | 0.5–0.75 | 2 |
| Nature factor | 600–1000 m | 3 | 25–35° | 3 | Grass land | 3 | 0.25–0.5 | 3 |
| | 1000–1400 m | 4 | 35–45° | 4 | Cultivated land | 4 | 0.1–0.25 | 4 |
| | 1400–2835 m | 5 | >45° | 5 | Building land and unused land | 5 | <0.1 | 5 |
| | River distance buffer zone (weight: 0.1332) | | Rail/highway buffer zone (weight: 0.1044) | | Grade I road buffer zone (weight: 0.0900) | | | |
| | 500–1000 m | 1 | >4000 m | 1 | >3000 m | 1 | | |
| | 1000–2000 m | 2 | 3000–4000 m | 2 | 2000–3000 m | 2 | | |
| Social factor | 2000–3000 m | 3 | 2000–3000 m | 3 | 1000–2000 m | 3 | | |
| | >3000 m | 4 | 1000–2000 m | 4 | 500–1000 m | 4 | | |
| | >4000 m | 5 | <1000 m | 5 | <500 m | 5 | | |

However, corridors between source points were too dense, and there were many similar or repeated corridors between source points that were not conducive to the planning and construction of the actual ecological corridors. Based on the idea of LCPs [49], this

study optimized a corridor under the condition of ensuring the interconnection between source points, protecting biodiversity and the flow of biological species. In addition, ecological nodes can enhance the overall ecological connectivity of the region and are located on the ecological corridor, which has an important impact on whether the ecological correlation can be established between source points. In terms of space, ecological nodes are ecological circulation hubs in the ecological network. This study takes the intersection of ecological corridors as the ecological node [15]. Figure 6b reports the spatial distribution of 114 ecological corridors and 28 ecological nodes identified. Ecological nodes were mainly distributed in Zhangjiakou City and Chengde City in northern Hebei Province. Southern cities had fewer nodes, some even none, and these urban landscapes were less ecologically connected and vulnerable to human activities. Combined with the distribution of ecological nodes and land use type, it can also be seen that the distribution of nodes had a certain distance from building land and cultivated land, indicating that nodes were less likely to be disturbed by human beings. From the distribution position of nodes, it can be seen that nodes may be affected by changes in land use types, such as land degradation and soil erosion. Therefore, in the future, the protection of nodes should mainly focus on restoring local landscape types.

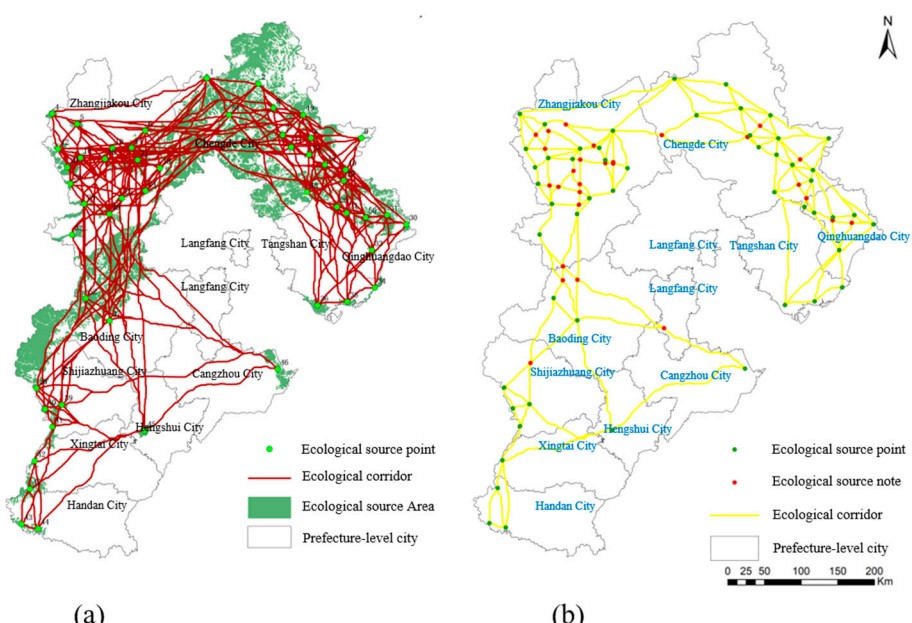

**Figure 6.** Spatial distribution of ecological network construction. (**a**) Spatial distribution of potential ecological corridors in Hebei Province. (**b**) Spatial distribution of ecological nodes in Hebei Province.

## 4. Discussion

Based on the spatial distribution analysis of ecological sources, it is evident that the majority of these sources are situated in the northern region of the study area. In contrast, the central and southern parts of Hebei Province are predominantly characterized by vast cultivated lands with fewer ecological patches. As Hebei Province is a significant agricultural province in China, it has the potential to leverage its local development characteristics to initiate agricultural transformations while simultaneously prioritizing ecological protection through the promotion of green agriculture.

Observing the current distribution of ecological corridors in the study area, it is apparent that the overall distribution is more concentrated in the north and less prevalent in the south. Firstly, it is essential to recognize that ecological sources and corridors serve as fundamental components of an ecological network. To enhance landscape connectivity, it may be beneficial to augment the number of ecological sources and expand the scope of ecological corridors. Secondly, the construction of actual ecological corridors necessitates a certain width, as narrow corridors can impede the exchange of materials and energy

between organisms and their environment. For areas vulnerable to ecological fragility, the establishment of corridor buffer zones could be considered to mitigate the pressures of environmental deterioration to a certain extent. Thirdly, it is important to note that large transportation infrastructures such as railways and highways can result in regional ecological fragmentation. However, these roads also play a crucial role in provincial economic development. As a solution, the implementation of green belts on both sides of such roads could help alleviate the impacts of ecological fragmentation on the corridors.

## 5. Conclusions

This study utilized land use data from Hebei Province to analyze the spatiotemporal changes in landscape patterns and their underlying causes. The landscape index was used to classify land use types into seven categories, identifying the core ecological patches within the study area. These core areas were then combined with local ecological protection areas to conduct landscape connectivity analysis. Subsequently, important ecological patches were identified, and seven resistance factors were selected to create a minimum resistance surface. The ecological network was constructed using the MCR model, extracting ecological corridors and nodes to analyze the structural characteristics and existing issues of the current ecological network. The research findings indicate that, during the period from 2000 to 2020, the main land use types in the study area were cultivated land, grass land, and wood land. Cultivated land showed a decline, while the change in grass land and wood land area mainly resulted from their transformation between one another. The landscape index results revealed that grass land, wood land, and building land had large PD values, indicating high landscape fragmentation, low aggregation, and weak connectivity. Cultivated land dominated the LPI value, making it the dominant landscape type in the study area. Grass land, wood land, building land, and cultivated land all exhibited higher shape scale index values, which indicated greater landscape complexity. Moreover, they all had higher LSI values, indicating irregular shape. Metrics such as SPLIT, SHDI, and SHEI showed increasing values, suggesting that regional landscape types gradually became more uniform in quantity and spatial distribution, ultimately enhancing landscape ecosystem stability. Additionally, based on the MSPA analysis, the landscape types were classified into seven categories in 2020. Core patch areas were predominantly distributed in the northern cities of Hebei Province. A total of 52 ecological patches with significant dPC values were identified as ecological source areas, and their spatial distribution showed a decreasing trend from north to south. Consequently, regional ecological security should be improved, taking into consideration local development conditions. Furthermore, seven resistance factors were selected, and a comprehensive resistance surface was constructed using their resistance values and weights. The potential ecological network was identified using the MCR model, comprehensive resistance surface, and ecological source calculations, resulting in the construction of 117 ecological corridors and the identification of 28 ecological nodes.

**Author Contributions:** Writing—review and editing, S.G.; writing—original draft, S.L.; formal analysis, S.W.; funding acquisition, B.T.; funding acquisition, Y.H.; data curation, M.C.; software, M.S. All authors have read and agreed to the published version of the manuscript.

**Funding:** This work was supported by Open Project of Key Laboratory of Geological Resources and Environment Monitoring and Protection in Hebei Province (grant number JCYKT202310), Shijiazhuang Science and Technology Plan Project (grant number 221790441), National Social Science Funding Project (grant number 20BJY047), 2022 Graduate Demonstration Courses in Hebei Province (grant number 15117544), and Major Science and Technology Special Fund of Hebei Normal University (grant number 13117079).

**Institutional Review Board Statement:** Not applicable.

**Informed Consent Statement:** Not applicable.

**Data Availability Statement:** Research data from this study will be made available on request (tianbing@hebtu.edu.cn, accessed on 20 August 2023).

**Conflicts of Interest:** The authors declare no conflict of interest.

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
