# Peer review of "Study on the Spatiotemporal Evolution of the Ecological Landscape and Construction of an Ecological Network: A Case Study of Hebei Province"

_sustainability, doi:10.3390/su152115661_

Round 1
Reviewer 1 Report
Comments and Suggestions for Authors
This paper calculates the landscape pattern index of each land use type and constructs the ecological network based on the three remote sensing images and other related data in 2000, 2010 and 2020 to explore the direct impact of land use change on landscape pattern and the indirect impact of ecological pattern. However, there are still some problems, and further revision is suggested to improve the quality of the manuscript.
(1) The manuscript lacks a discussion section.
(2) Line 209 mentions landscape type transfer, but the table is Land use type changes from 2000 to 2020, and does not analyze the dynamic changes in landscape patterns.
(3) The changes in PD, LPI, LSI, and AI from 2000-2020 are stated in lines 226-259, but the reasons for the changes are not analyzed or explored in depth. It is recommended that this section be clarified.
(4) In line 253 of the article, the AI value describes the aggregation degree of the patch and shows an in-253 verse relationship with the PD value, how does it prove that there is an inverse relationship between AI and PD? could the authors please explain.
(5) While selecting the RESISTANCE FACTORS. it is mentioned that Class I highway is selected as the resistance factor, there are various classifications of highway classes, why only Class I highway is selected as the resistance factor, authors are requested to clarify.
(6) Is the ecological network construction in the article 2000, 2010, or 2020? The previous article analyzes the land use type changes from 2000-2020, but in the ecological network construction, why is there no comparative analysis of the ecological network structure changes from 2000-2020.
Comments on the Quality of English Language
This paper calculates the landscape pattern index of each land use type and constructs the ecological network based on the three remote sensing images and other related data in 2000, 2010 and 2020 to explore the direct impact of land use change on landscape pattern and the indirect impact of ecological pattern. However, there are still some problems, and further revision is suggested to improve the quality of the manuscript.
(1) The manuscript lacks a discussion section.
(2) Line 209 mentions landscape type transfer, but the table is Land use type changes from 2000 to 2020, and does not analyze the dynamic changes in landscape patterns.
(3) The changes in PD, LPI, LSI, and AI from 2000-2020 are stated in lines 226-259, but the reasons for the changes are not analyzed or explored in depth. It is recommended that this section be clarified.
(4) In line 253 of the article, the AI value describes the aggregation degree of the patch and shows an in-253 verse relationship with the PD value, how does it prove that there is an inverse relationship between AI and PD? could the authors please explain.
(5) While selecting the RESISTANCE FACTORS. it is mentioned that Class I highway is selected as the resistance factor, there are various classifications of highway classes, why only Class I highway is selected as the resistance factor, authors are requested to clarify.
(6) Is the ecological network construction in the article 2000, 2010, or 2020? The previous article analyzes the land use type changes from 2000-2020, but in the ecological network construction, why is there no comparative analysis of the ecological network structure changes from 2000-2020.
Author Response
Thank you very much for your great efforts on our manuscript.
Comment 1: The manuscript lacks a discussion section.
Response: Thank you for your valuable advice. As Reviewer considered, we have added the Discussions section and optimized the Results section, which have marked in red.
Comment 2: Line 209 mentions landscape type transfer, but the table is Land use type changes from 2000 to 2020, and does not analyze the dynamic changes in landscape patterns.
Response: Thank you for your valuable advice. As Reviewer considered, the transfer matrix shows the change of land use type. In this study, 7 landscape indices were selected from the patch type scale and landscape type scale, and their dynamic changes were demonstrated from the aspects of landscape quantity, structure and form. We have also revised the Analysis method section to facilitate readers' better understanding.
Comment 3: The changes in PD, LPI, LSI, and AI from 2000-2020 are stated in lines 226-259, but the reasons for the changes are not analyzed or explored in depth. It is recommended that this section be clarified.
Response: Thank you for your valuable advice. As Reviewer considered, we added the PD, LPI, LSI and AI values change reason exposition. SPLIT, SHDI and SHEI values discussions were also added to this section, which will help readers understand better.
Comment 4: In line 253 of the article, the AI value describes the aggregation degree of the patch and shows an in-253 verse relationship with the PD value, how does it prove that there is an inverse relationship between AI and PD? could the authors please explain.
Response: Thank you for your valuable advice. As Reviewer considered, we are very sorry for the misunderstanding caused by Line 253. PD value reports the degree of differentiation of a landscape type. And AI value provides the degree of aggregation of the patches. There is no inverse relationship between the two. We have revised the relevant description and marked in red.
Comment 5: While selecting the RESISTANCE FACTORS. it is mentioned that Class I highway is selected as the resistance factor, there are various classifications of highway classes, why only Class I highway is selected as the resistance factor, authors are requested to clarify.
Response: Thank you for your valuable advice. As Reviewer considered, we have added relevant literature citations and marked in red. Higher-grade roads, including highways, grade â… roads and railways, have a significant impact on landscape-patch-corridor connectivity.
Comment 6: Is the ecological network construction in the article 2000, 2010, or 2020? The previous article analyzes the land use type changes from 2000-2020, but in the ecological network construction, why is there no comparative analysis of the ecological network structure changes from 2000-2020.
Response: Thank you for your valuable advice. As Reviewer considered, this study analyzed the spatial distribution of ecological source area and ecological network in Hebei Province in 2020 through MCR model. Focus on the assessment of the current ecological pattern and propose corresponding solutions in the discussion section. We have added a reference to the year in the description of this section and marked in red.
Reviewer 2 Report
Comments and Suggestions for Authors
Thanks for inviting me to review the manuscript titled "study on ecological landscape spatiotemporal evolution and ecological network construction : a case study of Hubei province". The paper appears to be interesting as it investigates the effect of land use change on ecological security pattern and ecological network was developed from the results. However, the authors didn’t review the wider literature on ecological network as they focused mainly on studies conducted in China. Hence, it is difficult to get sense of the broader contribution of the paper to knowledge. They need to review wider literature to get at this.
In addition, there are typographical errors in a number of places that need to be corrected.
My recommendation is a thorough review of the literature and a clear statement of the wider contribution of the study to knowledge.
Comments on the Quality of English Language
Needs some typographical error correction.
Author Response
Thank you very much for your great efforts on our manuscript.
Comment 1: The authors didn’t review the wider literature on ecological network as they focused mainly on studies conducted in China. Hence, it is difficult to get sense of the broader contribution of the paper to knowledge. They need to review wider literature to get at this.
Response: Thank you for your valuable advice. As Reviewer considered, we added some references to relevant literature and revised existing literature.
Comment 2: There are typographical errors in a number of places that need to be corrected.
Response: Thank you for your valuable advice. As Reviewer considered, we have revised typographical errors in the manuscript.
Comment 3: A thorough review of the literature and a clear statement of the wider contribution of the study to knowledge.
Response: Thank you for your valuable advice. As Reviewer considered, we have further studied the relevant literature, and have added Discussion section to explain the application of the results.
Round 2
Reviewer 2 Report
Comments and Suggestions for Authors
The authors attempt to improve the manuscript, but can still be improved in terms of an argument the paper makes at the introduction. After this statement
"Research results will show theoretical and technical basis for ecosystem protection and rational development and utilization of resources in Hebei Province and other provinces and provide reference for the evolution of land use types and construction of ecological security pattern."
The authors need to:
State briefly the overall argument from these findings, that is the "theoretical and technical basis for ecosystem protection and rational development and utilization of resources".
What is theoretical and practical argument in this regard?
What is the argument about the "evolution of land use types and construction of ecological security pattern"?
How are all these related?
Answer all these questions in one paragraph so that I can assess your contribution more clearly.
Comments on the Quality of English Language
Only proofreading is required.
Author Response
Thank you very much for your great efforts on our manuscript.
Comment 1: The authors attempt to improve the manuscript, but can still be improved in terms of an argument the paper makes at the introduction. After this statement "Research results will show theoretical and technical basis for ecosystem protection and rational development and utilization of resources in Hebei Province and other provinces and provide reference for the evolution of land use types and construction of ecological security pattern."
Response: Thank you for your valuable advice. As Reviewer considered, we have modified this section and marked it in red.
Comment 2: The authors need to: State briefly the overall argument from these findings, that is the "theoretical and technical basis for ecosystem protection and rational development and utilization of resources". What is theoretical and practical argument in this regard? What is the argument about the "evolution of land use types and construction of ecological security pattern"? How are all these related?
Response: Thank you for your valuable advice. As Reviewer considered, this study demonstrates the necessity of ecosystem conservation and sustainable resource development from both theoretical and practical perspectives. Protecting ecosystems, maintaining biodiversity, and ensuring climate stability are all based on ecological principles. At the same time, practical resource management approaches, sustainable agricultural development, green technologies, and natural capital accounting provide viable pathways to balance resource utilization and ecosystem conservation, promoting sustainability for both current and future generations.
There is a close relationship between the evolution of land use types and the construction of ecological security patterns. Changes in land use types can directly impact the ecological security pattern, while the construction of ecological security patterns can guide the rational planning and management of land use to some extent. Their coordination and rational planning can contribute to maintaining ecosystem stability, biodiversity, and enhancing societal ecological security. Therefore, when planning land use and designing ecological security patterns, it is necessary to consider their relationship comprehensively, in order to achieve sustainable land use and ecosystem management.
Round 3
Reviewer 2 Report
Comments and Suggestions for Authors
Thank you for responding to the queries. I do not have further comments.